# Where2Explore: Few-shot Affordance Learning for Unseen Novel Categories of Articulated Objects

**Chuanruo Ning**[1,3]    **Ruihai Wu**[2,3,5]    **Haoran Lu**[1,3]    **Kaichun Mo** [4]    **Hao Dong** [3,5*]

[1]School of EECS, PKU    [2]School of CS, PKU    [3]CFCS, School of CS, PKU    [4]NVIDIA
[5]National Key Laboratory for Multimedia Information Processing, School of CS, PKU

## Abstract

Articulated object manipulation is a fundamental yet challenging task in robotics. Due to significant geometric and semantic variations across object categories, previous manipulation models struggle to generalize to novel categories. Few-shot learning is a promising solution for alleviating this issue by allowing robots to perform a few interactions with unseen objects. However, extant approaches often necessitate costly and inefficient test-time interactions with each unseen instance. Recognizing this limitation, we observe that despite their distinct shapes, different categories often share similar local geometries essential for manipulation, such as pullable handles and graspable edges - a factor typically underutilized in previous few-shot learning works. To harness this commonality, we introduce 'Where2Explore', an affordance learning framework that effectively explores novel categories with minimal interactions on a limited number of instances. Our framework explicitly estimates the geometric similarity across different categories, identifying local areas that differ from shapes in the training categories for efficient exploration while concurrently transferring affordance knowledge to similar parts of the objects. Extensive experiments in simulated and real-world environments demonstrate our framework's capacity for efficient few-shot exploration and generalization.

## 1 Introduction

Articulated objects, such as doors, drawers, scissors, and faucets, are ubiquitous in our daily lives. Therefore, the ability of robots to manipulate these objects is of critical importance. Many previous works have been done on perceiving and manipulating articulated objects [25, 35, 38, 3]. However, due to the significant variance in the objects' structure, 3D geometry, and articulation types across categories, developing efficient perception and manipulation systems that can generalize to those variations remains challenging [7, 25, 41].

An intuitive solution to equip models with generalized manipulation knowledge is training them on large-scale datasets. However, conducting real-world interactions with diverse objects or acquiring 3D models encompassing potential categories can be prohibitively time-consuming and costly. Moreover, this approach could still fail with the emergence of new object categories or designs (*e.g.*, a cup with novel geometries resembling a gourd as shown in Figure 1).

Since encountering novel objects is inevitable in real-world applications, few-shot learning, which allows robots to propose interactions with novel objects and adapt their understanding to them, has emerged as a promising solution. However, previous few-shot learning works for articulated object manipulation usually focus on instance-level exploration and adaptation, requiring test-time interactions on each novel object [35, 27]. This limitation hinders the efficiency and safety of real-world applications of robots.

This paper investigates the open question of cross-category few-shot learning, in which the model is required to understand how to manipulate a novel category via interactions with a limited number of objects. After the few-shot exploration, the model should be able to manipulate other unseen objects within the same category without further test-time interactions.

---

[*]Corresponding author.

37th Conference on Neural Information Processing Systems (NeurIPS 2023).

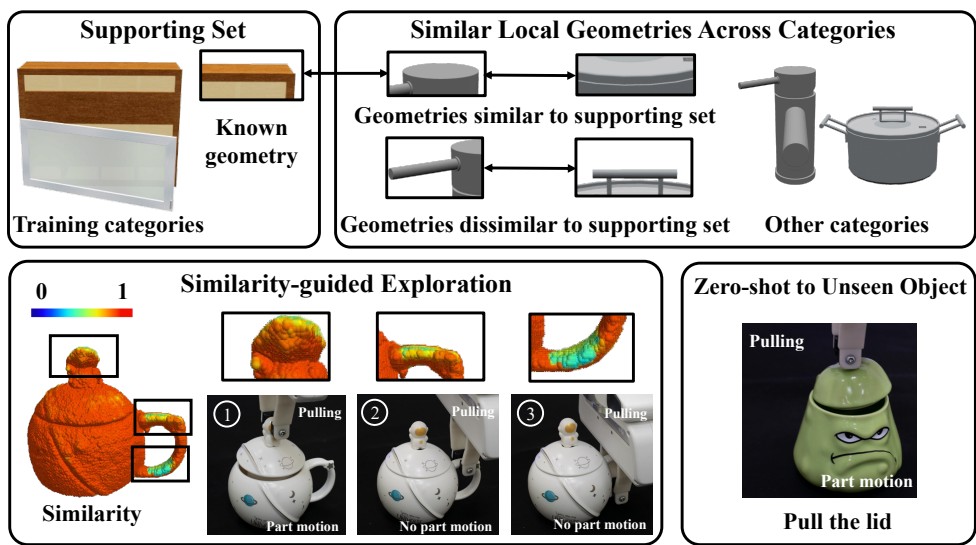

Figure 1: **Where2Explore framework.** Our model, solely trained on training categories (Top Left) and having never seen mugs, utilizes the underlying similarity in local geometries across different categories (Top Right), enabling it to identify uncertain yet important areas for interaction (Bottom Left). After minimal interactions, our model could manipulate unseen objects (Bottom Right) in this category.

Different from instance-level few-shot learning that focuses on discovering kinematic and dynamic information of a specific object, cross-category few-shot learning proposes a more demanding requirement for the exploration strategy to select informative interactions on different object categories. Considering the substantial semantic and geometric gap between known shapes and novel categories, forming an efficient exploration strategy for out-of-distribution objects is challenging. However, we point out that despite distinct overall shapes, different categories often share similar local geometries crucial for manipulation (*e.g.*, pullable handles, pushable boards, and graspable edges, as shown in Figure 1). This property of possessing similar significant geometries across different categories is typically ignored or underutilized by previous instance-level few-shot learning studies.

To effectively discover these critical local geometries for cross-category few-shot learning, we propose a 'Where2Explore' framework by explicitly requesting the system to estimate the semantic similarity of geometries on novel objects with geometries already known by the model (As shown in the top part of Figure 1). When faced with objects from a novel object category (*e.g.*, mugs), our framework identifies the uncertain yet important areas on the novel objects to interact with (Bottom Left of the Figure). Via fine-tuning our network with the interactions on novel objects, the model could generalize to unseen objects within this novel category (Bottom Right).

Our system is built in several steps. Firstly, we require a representation that encapsulates diverse semantic and geometric information from known categories to form a supporting set from which we could expand the learned knowledge to novel categories. The desired properties in our system are fulfilled by point-level affordance, which provides per-point manipulation priors with detailed semantic and geometric information on various objects [25, 38, 35, 46]. Next, we implement a 'similarity network' to measure the geometric similarity between shapes in different categories and those in the supporting set. This is achieved by partitioning the training set, exposing our similarity module to a wider range of categories, and supervising the learning of similarity in a cross-category manner. Finally, we perform few-shot learning on novel categories by proposing interactions on the low-similarity areas, indicating unseen yet significant geometries.

We evaluate our framework by training our model on constrained object categories and applying few-shot learning to novel categories with limited shapes. After the exploration, we evaluate our model on unseen objects in the novel categories. The results demonstrate our framework's capability to efficiently explore novel categories by exploiting geometric similarity. Additionally, we examine our framework's robustness across diverse combinations of training and testing categories, yielding consistent results.

In summary, the contributions of this paper include:

- Exploring the challenging task of cross-category few-shot learning for articulated object manipulation, requiring the model to capture fine-grained geometric information from an entirely new category using a few interactions with limited instances.

- Introducing the 'Where2Explore' affordance learning framework that explicitly measures the semantic similarity of local geometries across different categories, which successfully guides the exploration on novel categories with only a few interactions.

- Our experiments, both in a simulator and the real world, show that our proposed framework can efficiently explore novel categories and generalize to unseen instances.

## 2    Related Work

**Perceiving Articulated Objects for Manipulation.**    Future home-assistant robots need to possess the ability to perceive and manipulate a wide variety of articulated objects within human environments. Previous studies have made significant advancements in estimating and tracking the segmentation of the articulated parts [18, 43, 8, 15, 33], the 6-DoF part poses [20, 21, 36, 22, 9], the joint parameters [32, 34, 45, 13], and digital twins [26, 16, 12, 14]. Based on these perceptual signals, subsequent motion planners and controllers [31, 4, 2, 24, 1] can manipulate the articulated objects. Our work focuses on learning dense manipulation affordance heatmaps over the input 3D object, augmenting the basic part and joint parameters with fine-grained geometry-to-action mappings.

**Affordance Learning on Articulated Objects.**    The concept of affordance [11] plays a pivotal role in facilitating the manipulation of articulated objects. In the literature, researchers have been exploring learning manipulation affordance on articulated objects [19, 17, 23, 6, 25, 38, 41, 30, 10, 42, 40, 5]. Although these works can accurately predict manipulation affordances for articulated objects, they frequently encounter difficulties handling novel unseen objects significantly different from the training data, which is the core problem we study in this paper. A noteworthy piece of research closely aligned with ours is the AdaAfford system [35] that learns to fine-tune the affordance prediction of novel objects using a limited number of interactions. While AdaAfford focuses on rapidly adapting to a single novel instance within known training object categories, our work addresses the more arduous challenge of generalizing to an entirely new category of articulated objects.

## 3    Problem Formulation

In line with prior affordance learning studies [25, 38, 46, 37], given an N-point 3D partial point cloud observation of an articulated object $O \in \mathbb{R}^{N \times 3}$ and a set of action directions and gripper orientations $\{R_1^p, R_2^p, R_3^p, \cdots | R_i^p \in SO(3)\}$ on each point, the visual manipulation affordance is defined as a dense prediction $Aff \in \mathbb{R}^{N \times 3}$, where $a \in [0, 1]^N$ on each point indicates whether the action on that point would result in a part motion. This definition forms a fine-grained geometry-to-action mapping.

During the few-shot exploration, the model needs to propose a few interactions $I = \{I_1, I_2, \cdots\}$ sequentially. Each interaction $I_i = (O_i, p_i, R_i)$ represents a task-specific hard-coded trajectory defined in Where2Act [25], parametrized by the interaction point $p_i \in O_i$ and the action direction and gripper orientation $R_i \in SO(3)$. The interaction will result in a part motion $m_i$. The model will be fine-tuned by the proposed interactions $I$ and corresponding outcomes $\{m_1, m_2, \cdots\}$.

## 4    Method

As shown in Figure 2, we propose the 'Where2Explore' framework to explicitly leverage the similar semantics on local geometries shared across different categories for cross-category few-shot exploration. To achieve this, we divide the training categories into two parts - the affordance category $C_{aff}$ and the similarity categories $C_{sim}$. We begin by learning point-level manipulation affordance on the affordance category to create a supporting set containing semantic and geometric information on known shapes (Figure 2, Left) 4.1. Next, we introduce the 'similarity module' to form a representation that connects the geometries in the supporting set with geometries across category boundaries. This module is trained by comparisons between the geometries in $C_{aff}$ with those in $C_{sim}$. (Middle) 4.2. Then, to expand the supporting set along the similarity representation we built, we perform few-shot learning on novel categories with the guidance of the similarity module, which transfers its prediction to novel categories through shared local geometries (Right) 4.3. Finally, we describe the network and training strategy 4.4.

### 4.1    Affordance Learning for Building Supporting Set

To conduct the cross-category few-shot exploration task, firstly, we need to build a supporting set from which we can expand our knowledge to a broader range of categories. This requires a representation that encapsulates the learned semantic and geometric information in known categories.

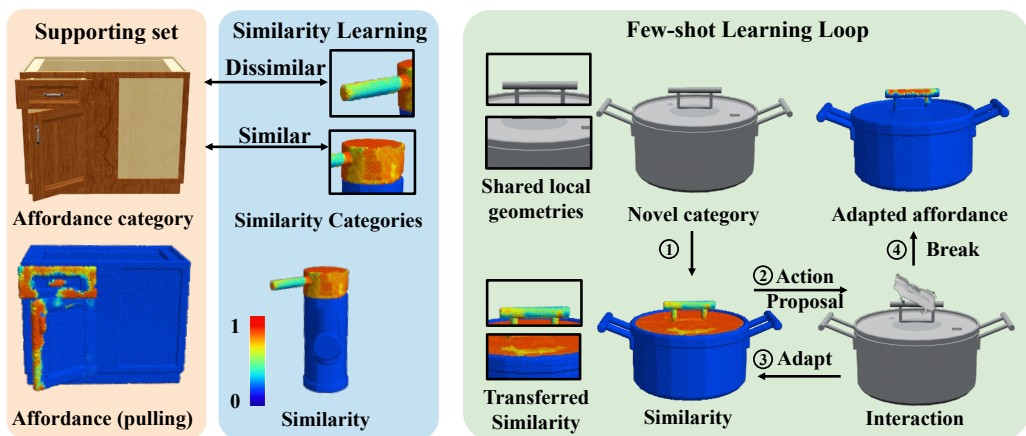

Figure 2: Method overview: We first employ affordance learning on the affordance category to form our supporting set (Left). Then, we estimate the semantic similarity between learned geometries and geometries from the similarity categories (Middle). Finally, leveraging the shared local geometries in novel categories, we conduct few-shot learning with the transferred similarity prediction (Right).

Visual manipulation affordance has been proven by previous works to have fine-grained manipulation information and is able to generalize to unseen objects within the same category [25, 38, 35]. We exploit these valuable properties to build up our supporting set. Following Where2Act [25], we build a module to predict the per-point affordance. Given a specific action $R_i$ on a point $p_i$ of a partial point cloud $O_i$, the affordance module is required to predict whether the given action will result in a part motion. We train the affordance module on training categories to build up its semantic understanding of geometries in the training categories.

However, previous affordance learning works suffer a dramatic performance drop when tested on novel categories. To enable our framework to smoothly expand the supporting set to include wider object categories, we need a cross-category representation that links similar geometries across categories.

## 4.2 Cross-category Similarity Learning

We propose to explicitly estimate the similarity between geometries from different categories with the learned geometries in the supporting set. The similarity should act as a set of 'bridges' connecting shapes in the supporting set and the geometries from different categories. To be specific, the proposed similarity should be equipped with the following properties. Firstly, the similarity should be conditioned on specific actions (the action type, action direction, and gripper orientation) because even geometrically similar areas can have distinct semantic meanings when the action is different. For example, handles are significant in pulling whereas less important in pushing, and a horizontal handle could not be grasped by a gripper whose pose is also horizontal. Besides, similarity should be based on the current knowledge of affordance, showing high similarity when the learned affordance could directly generalize to a given geometry, which indicates this geometry is semantically similar to the supporting set and verse versa.

To achieve the first property, as shown in the middle of Figure 3, we propose a 'similarity module' to predict the semantic similarity. The similarity module is designed to take a partial point cloud of an object $O_i \in \mathbb{R}^{3 \times N}$, a set of action directions and gripper orientations $\{R_i\}$ on each point, and is required to predict per-point similarity $Sim \in \mathbb{R}^N$ based on these inputs. This design enables the similarity to be aware of specific actions and thus has the potential to contain manipulation semantics instead of only relying on the shape geometry.

In order to acquire the second property, the key is to expose our similarity module to categories that are broader than what our affordance module is trained on. As shown in the 3 (Left), we divide the training categories into two parts - the affordance category $C_{aff}$ that only contains one category and the similarity categories $C_{sim}$ that contains three categories (which also contains the affordance category). We train our affordance on $C_{aff}$ and supervise the similarity module using $C_{sim}$.

During training, the interactions on the affordance category (*e.g.*, cabinets) are used to supervise the affordance module, as shown in the previous section. However, interactions on similarity categories

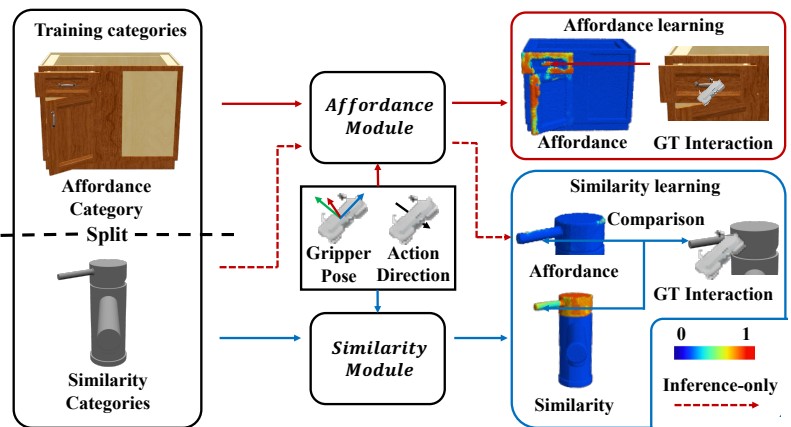

Figure 3: **Cross-category similarity learning.** We use a similarity module to predict the similarity conditioned on specific actions (Middle). While the affordance category is used to train the affordance representation (Top Right), the split similarity categories are used to supervise the learning of the similarity module by comparing the GT interactions with the affordance prediction (Bottom Right).

(*e.g.*, faucets, and windows) are kept from updating the affordance module. Instead, we compare the affordance prediction on objects from similarity categories with the ground-truth interaction results and use the accuracy of affordance prediction during training to supervise the learning of the similarity module, which is:

$$Sim(O_i, p_i, R_i) := Accu(Aff(O_i, p_i, R_i), m_i), O_i \in C_{sim}, \tag{1}$$

where $Sim$ and $Aff$ stand for the similarity and affordance prediction given a specific action. The $Accu$ is computed using the accuracy in predicting the affordance score of an action during training.

As straightforward as this approach may appear, the learned similarity holds valuable properties for cross-category exploration. Since the similarity is trained in a cross-category manner, it forces the network to focus on local geometries shared across different objects, which is beneficial for generalizing to novel categories. Moreover, the learned similarity is based on the current knowledge of affordance, highlighting the areas where the current model is uncertain while saving the unnecessary exploration on geometries that the affordance could directly generalize to. Finally, similarity defined through this approach could reveal the semantic meaning of geometries with regards to manipulating them instead of only relying on the similarity of shapes.

### 4.3 Few-shot Learning Loop

After the similarity linking the supporting set and different geometries are learned, we perform a few-shot exploration of novel categories (unseen by both modules) with the learned similarity module.

As shown in the right part of figure 2, when faced with a novel category, our framework will first predict the similarity of the objects. Thanks to the property that similarity is conditioned on action directions and gripper orientations, we could sample interactions in diverse directions and poses $R_i$ to indicate which action on what geometry $p_i \in O_i$ contains the most informative semantics. Then, by choosing the action with the lowest similarity prediction, the model performs a short-term manipulation trajectory and observes the result of the interaction as a part motion $m_i$ (the right part of the figure). Finally, both the affordance module and the similarity module will be updated by this interaction $(O_i, p_i, R_i, m_i)$ and be ready for the next prediction on the object to indicate where to explore. This loop will break if the similarity on the instance reaches a bar or the interaction budget is reached.

Through just a few interactions on the instances from an unseen category, our model explores the semantic and geometric significant areas of a novel category and captures the common features shared with already learned categories. Via these few-shot explorations, the knowledge of manipulation affordance on training categories is transferred to novel categories. Thanks to the generalization ability of affordance within one category, our adapted affordance could manipulate unseen objects from this category without additional interactions.

## 4.4 Network Architecture and Training Strategy

Our network consists of two modules - the affordance module and the similarity module. We use a PointNet++ segmentation network [29] encoder for extracting features from 3D partial point clouds. The encoder will output a per-point feature of 128 dimensions. We let the two modules share the same encoder since we want the similarity module to be based on affordance prediction. We employ Multilayer Perceptrons (MLP) with one hidden layer of size 128 to implement both decoders.

**Affordance Loss.** To supervise the learning of the affordance network, We deploy a binary cross-entropy loss, which measures the error between the affordance prediction of a given interaction and a binary label indicating whether the action results in a part motion (*i.e.*, $m_i$ reaches the threshold of 0.01).

**Similarity Loss.** To train the similarity module, we use an $\mathcal{L}_1$ loss to measure the distance between Similarity prediction and the ground truth accuracy. This accuracy is calculated as the ratio of correct predictions to total prediction attempts for each interaction during training.

We balance the positive and negative interactions during training and sampling the same amount of instances from different object categories equally. We train both modules simultaneously to learn the proposed affordance and similarity until they converge. Please see supplementary for more details.

## 5 Experiments

To demonstrate the ability of our framework to propose informative interactions for cross-category exploration efficiently. We intentionally train our model with constrained training categories. Then, we perform few-shot learning on a wide variety of categories using only a few instances. Finally, we test our fine-tuned model on unseen instances in novel categories to demonstrate that our model learns the general semantic and geometric information. We set up three baselines for comparisons. We also conduct ablation studies to prove the efficiency of our exploration strategy.

### 5.1 Data and Settings

**Data.** Following [25, 35], we use SAPIEN [39] with NVIDIA PhysX [28] as our simulator. We use 942 articulated 3D objects covering 14 categories to show our cross-category exploration ability.

To simulate the challenging cross-category few-shot task in the real world. We divide the categories into **only three training categories** and **11 novel categories**. The shapes in all the categories are further divided into two disjoint sets of training and testing shapes. See supplementary for more details.

**Experiment Settings.** We conduct experiments under two different manipulation action types (pushing and pulling). We first train the networks on three training categories and then perform few-shot learning on 11 novel categories. Finally, we evaluate the model by testing objects from the novel categories.

For the training stage, to filter out randomness and prove the universal effectiveness of our framework, we conduct experiments using **4 different training category combinations**, which are {cabinet, faucet, window}, {cabinet, switch, refrigerator}, {table, Faucet, refrigerator} and {table, switch, window}. These category combinations are chosen because they could cover representative articulations (revolute and prismatic joints). For baselines, we train the models using all training objects in training categories, whereas we divide the training categories into two parts to train our framework, as mentioned in the method. The first category in each combination is the training category.

For few-shot learning, we perform exploration on objects from the novel categories (*i.e.*, the rest 11 categories). We conduct two experiments. The first experiment is few-shot learning on all novel categories simultaneously, which requires the model to choose the objects and interactions that contain uncertain yet significant semantic properties. We also perform few-shot learning on each novel category separately to match the real-world scenario.

It is worth noticing that we only choose **10 instances from each novel category for few-shot learning** and evaluate the model on unseen objects in the novel categories, which challenges the model's ability to explore wisely in order to learn the semantic and geometric information that could generalize to unseen objects.

**Environment Settings.** Following Where2Act and AdaAfford [25, 35], we abstract away the robot arm and only use a Franka Panda flying gripper as the robot actuator. The input partial point cloud is assumed to be cleanly segmented out. To generate the point clouds, we mount an RGB-D camera with known intrinsic parameters 5-unit-length away pointing to the center of the target object.

## 5.2 Baselines, Ablations, and Metrics

**Baselines and Ablations.** We compare our framework with several baselines:

- **Where2Act** [25]: an affordance learning framework predicting the visual actionable affordance using a partial point cloud. During few-shot learning, Where2Act will sample interaction on the points with an affordance score closest to 0.5.

- **AdaAfford** [35]: an affordance learning method that explores test-time few-shot adaptation. During few-shot exploration, the interactions are proposed by a curiosity module that is optimized for discovering the dynamic information of a specific object.

- **PointEncoder** [44]: a pre-trained transformer framework that takes point cloud as input and could perform few-shot learning on classification and segmentation tasks. This baseline uses the pre-trained transformer encoder to extract features for few-shot affordance learning.

We select **Where2Act** as a baseline to compare the exploration ability of certainty represented by our similarity with the certainty defined in classification tasks. We use **AdaAfford** to evaluate the ability of instance-level exploration strategy on cross-category few-shot learning. We select **PointEncoder** to compare our framework with a network pre-trained on large-scale datasets.

Besides, we compare to ablated versions of our method to verify our exploration strategy:

- **No-explore (lower bound)**: our affordance model directly evaluated on novel categories without few-shot exploration, which represents the lower bound of few-shot learning.

- **Explore-random**: a variant of our proposed framework that explores novel objects through random interactions.

- **Explore-noSim**: a variant of our proposed framework that uses the same exploration strategy as Where2act instead of using the similarity module.

- **Full-data (upper bound)**: our affordance model trained on all categories with abundant data. We choose this ablation to represent the upper bound of affordance learning.

**Evaluation Metrics.** Following Where2Act [25] and AdaAfford [35], we use the F-score, balancing the precision and recall, to evaluate the predictions of the visual affordance and use the sample successful rate to evaluate the ability of the learned affordance to propose successful actions. We calculate the sample success rate by randomly selecting one action predicted as successful by the affordance module, performing the interaction, and observing the result. The final rate is reported as the percentage of successful interactions in the simulation. For both the F-score and sample success rate, we use the average score of the four different training category combinations.

## 5.3 Quantitative Results and Analysis

| Method | F-score | | Sample successful rate | |
| --- | --- | --- | --- | --- |
| | Pushing | Pulling | Pushing | Pulling |
| Where2Act | 25.6 / 28.0 / 30.4 | 6.4 / 7.5 / 8.5 | 15.7 / 17.0 / 19.9 | 3.9 / 4.3 / 6.2 |
| AdaAfford | 27.5 / 29.7 / 32.0 | 3.7 / 4.0 / 4.4 | 27.2 / 31.3 / 37.1 | 9.1 / 9.4 / 11.1 |
| PointEncoder | 19.4 / 19.4 / 29.9 | 2.9 / 4.6 / 5.9 | 11.6 / 10.9 / 29.9 | 1.8 / 3.1 / 9.2 |
| Ours | **35.4 / 38.5 / 41.6** | **12.1 / 12.5 / 24.2** | **31.3 / 37.3 / 39.5** | **11.5 / 13.4 / 14.9** |

Table 1: Few-shot learning on novel categories using different interaction budget (1, 2, 5).

Table 1 shows the results of few-shot learning on novel categories using different interaction budgets. Our method outperforms others in all metrics, particularly in 'pulling' actions. The pulling action is more demanding in selecting informative geometries to interact with (*e.g.*, drawer handles, kettle lids). Specifically, compared to **Where2Act**, our framework could more efficiently explore novel categories, proving that the cross-category similarity is a better representation of certainty on novel geometries than certainty defined in classification tasks. Compared to the **AdaAfford**, our results suggest that instance-level exploration strategies which focus on dynamic information for a single object fail to generalize well across categories. **Notably**, AdaAfford requires test-time interaction for affordance prediction, making the comparison unfair since our model predicts affordance from only visual observation. Compared with **PointEncoder**, we show that our framework better understands the semantic information for manipulation than a pre-trained encoder, even if it is trained on a large-scale dataset and achieves generalization ability on several tasks.

| Methods | Pushing unseen instances in novel categories | | | | | | | Pulling unseen instances in novel categories | | | | | | |
|---|---|---|---|---|---|---|---|---|---|---|---|---|---|---|
| | 🗑 | 🪣 | ▥ | 🚪 | ▦ | ▯ | 🖥 | 🗑 | 🪣 | ▥ | 🚰 | ▥ | ▯ | 🖥 |
| Where2Act | 22.1 | 10.5 | 42.8 | 43.4 | 31.2 | 47.4 | 51.7 | 8.9 | 6.0 | 13.1 | 12.1 | 2.5 | 5.4 | 8.3 |
| AdaAfford | 24.4 | 7.5 | 50.1 | **48.8** | 25.5 | 44.3 | 52.2 | 9.2 | 4.3 | 14.0 | 11.3 | 2.7 | 7.8 | 9.2 |
| PointEncoder | 20.4 | 14.2 | 29.3 | 24.1 | 22.7 | 26.8 | 29.8 | 3.9 | 9.6 | 7.7 | 7.8 | 4.7 | 8.9 | 9.0 |
| Ours | **36.5** | **15.6** | **60.5** | 48.5 | **39.7** | **61.5** | **66.0** | **26.6** | **15.8** | **28.8** | **19.1** | **8.7** | **16.4** | **13.8** |
| | F-score (%) | | | | | | | | | | | | | |
| Where2Act | 14.1 | 5.9 | 42.4 | 35.7 | 22.2 | 34.8 | 39.4 | 7.4 | 5.3 | 18.2 | 18.2 | 1.5 | 3.0 | 4.5 |
| AdaAfford | 14.4 | 7.5 | 47.1 | **47.4** | 24.2 | 40.4 | 43.2 | 7.7 | 7.5 | 25.0 | 11.3 | 1.3 | 3.1 | 5.4 |
| PointEncoder | 13.1 | 3.4 | 18.7 | 17.3 | 12.4 | 17.5 | 21.0 | 3.0 | 3.9 | 4.3 | 7.8 | 0.6 | 4.3 | 3.6 |
| Ours | **29.5** | **9.6** | **54.5** | 41.9 | **32.8** | **49.2** | **54.7** | **17.1** | **16.0** | **35.5** | **21.4** | **15.1** | **11.3** | **15.4** |
| | Sample successful rate (%) | | | | | | | | | | | | | |

Table 2: Evaluation of few-shot learning on different categories separately (5 interaction budget)

| Method | F-score | | Sample successful rate | |
|---|---|---|---|---|
| | Pushing | Pulling | Pushing | Pulling |
| No-explore | 20.6 | 3.9 | 12.1 | 3.2 |
| Explore-random | 24.9 / 28.8 / 29.0 | 4.0 / 6.1 / 9.2 | 15.0 / 18.4 / 21.6 | 3.4 / 5.2 / 5.1 |
| Explore-noSim | 24.8 / 25.2 / 30.2 | 6.8 / 6.7 / 8.4 | 15.0 / 19.2 / 25.5 | 5.4 / 6.2 / 5.3 |
| Ours | **35.4 / 38.5 / 41.6** | **12.1 / 12.5 / 24.2** | **31.3 / 37.3 / 39.5** | **11.5 / 13.4 / 14.9** |
| Full-data | 47.9 | 27.1 | 40.3 | 13.9 |

Table 3: Ablations on the exploration strategy using different interaction budget (1, 2, 5).

We also conduct few-shot affordance learning on representative categories separately to match the real-world scenario. Table 2 presents the quantitative comparisons against the baselines showing that our method achieves the best performance in most entries (especially in pulling tasks).

In Table 3, we compare our full method against several variants of our framework. Compared with **No-explore**, we observe that the proposed similarity could smoothly guide the cross-category generalization of affordance. Compared with other exploration strategies **Explore-random** and **Explore-noSim** that fail to discover important local areas, our strategy is dramatically more effective and efficient. Our framework also achieves comparable performance compared with **Full-data**, which is trained on all categories with abundant data. Considering that our framework only uses 0.3% of the original data, it further proves the efficiency of our exploration strategy.

## 5.4 Qualitative Results and Analysis

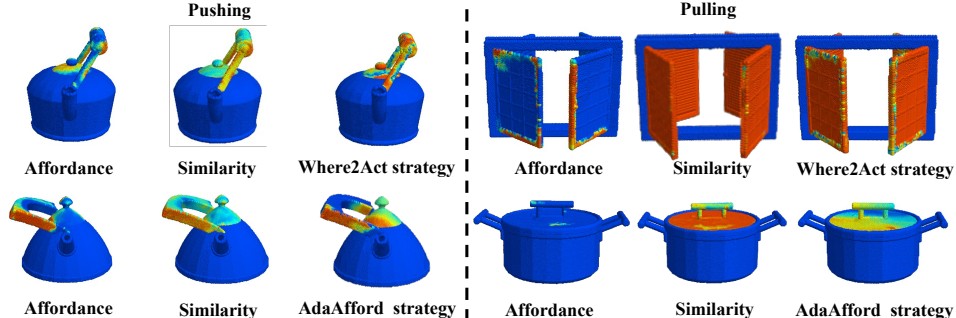

Figure 4: Visualization of different exploration strategies on novel objects. The action directions are set to the normal direction of each point in this visualization.

Figure 4 compares the visualization of our proposed similarity with exploration strategies of Where2Act [25] and AdaAfford [35] on novel categories. Compared with **Where2Act**, our proposed similarity is more geometric-aware, successfully discovering uncertain geometries (*e.g.*, whether the handle joint is at its limit) to interact with and familiar shapes (*e.g.*, windows) to save exploration budget. Compared with **AdaAfford**, which fails to generalize to novel categories, our framework could still propose reasonable exploration strategies on novel categories leveraging local similarity.

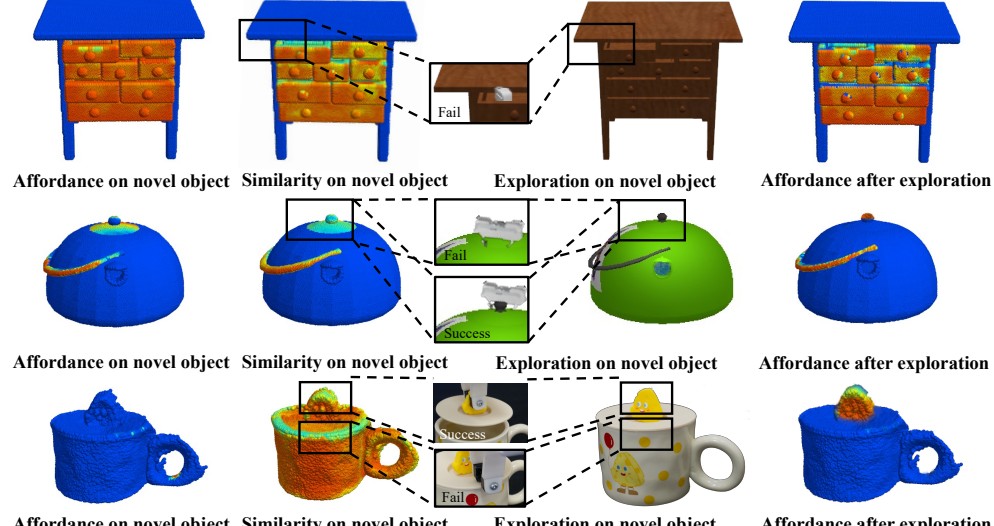

Figure 5: Pushing (top) and pulling (middle and bottom) affordance and similarity prediction on novel object categories. Although Affordance fails to directly generalize to novel categories (Left) via interacting on low-similarity areas (Middle), our framework could learn the semantic information on them (Right). In this visualization, for objects in the simulator, the action directions are set to the normal direction of each point. For objects from the real world, the direction is the vertically up direction.

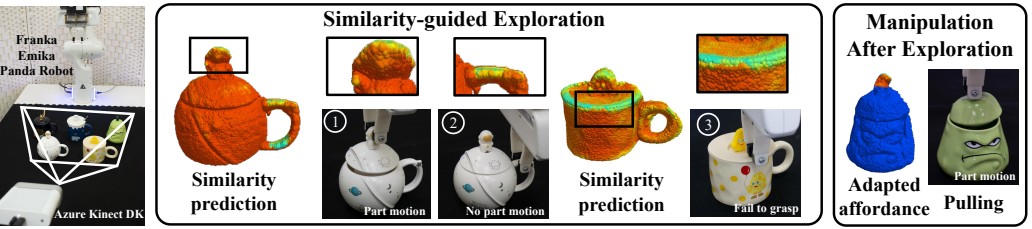

Figure 6: Real-world experiment set up (Left), similarity-guided exploration (Middle), and manipulation after exploration (Right). The pulling direction is vertical in this visualization.

Figure 5 shows the visualization of our proposed affordance and similarity on novel categories. While affordance fails to directly generalize to novel objects (Left), the similarity module can still discover areas that contain uncertain yet important semantic information to interact with (Middle). With a few similarity-guided interactions on the novel category, the affordance could capture the geometric information of novel objects. **Note that** although we choose the same novel object for visualization, the adapted manipulation affordance could directly generalize to other unseen objects in this category.

Figure 6 shows the qualitative results in the real world. We require our model, which is only trained on cabinets, windows, and faucets, to perform a few-shot exploration on four mugs and manipulate another mug after the exploration. We use pulling as our action primitive since it's more challenging. All objects are fixed to the tabletop to ensure the observed motion is a part motion instead of an entire movement. Please refer to the supplementary for more details.

## 6 Conclusion

We investigate the critical yet challenging task of cross-category few-shot affordance learning. The proposed 'Where2Explore' framework leverages the similarities in geometries across different categories to guide exploratory interactions on uncertain yet significant areas during few-shot learning. This study of cross-category few-shot exploration is beneficial to the real-world application of robots by empowering them to understand and manipulate novel categories through minimal interactions.

**Limitations and Future Works.** Our performance drops on categories with significant shape variance. Future works could use 'similarity' to divide local geometries into parts according to their manipulation properties, which might improve the generalization ability towards more objects.

**Ethics Statement.** Our work can empower future robots with the ability to explore and understand unseen objects. We do not see our work has any particular major harm or issue.

## 7 Acknowledgment

This work was supported by National Natural Science Foundation of China - General Program (62376006) and The National Youth Talent Support Program (8200800081).

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
