# OpenReview forum: "Where2Explore: Few-shot Affordance Learning for Unseen Novel Categories of Articulated Objects"
_NeurIPS.cc/2023/Conference — NeurIPS 2023 poster_

### Official Review · Reviewer_3SiD · 2023-06-12

**Soundness:** 2 fair
**Presentation:** 3 good
**Contribution:** 2 fair
**Rating:** 6
**Confidence:** 4

**Summary:**

This paper first proposes a new task, cross-category few-shot learning for articulated object manipulation. And it further proposes a framework that first explictly measures the semantic similarity of local geometries,  interacts with the objects of novel categories according to the similarity, uses the interaction results to update the affordance estimation network, and finally manipulate the target object. Authors perform experiments in simulation and the proposed method outperforms existing methods. They show a successful demo on a real robot.

**Strengths:**

The strengths of the paper are:
1. The paper formulates a new problem, cross-category few-shot affornace learning, which may be useful for robot applications;
2. The paper proposes a framework to address the proposed problem, which leverages the geometric similarity to guide the exploration on objects of novel categories. The paper designs a cross-category similarity learning pipeline to learn the similarity metric.

**Weaknesses:**

1. Statistics, eg. std, are not provided in the experimental results, especially the variance casued by the choose of few-shot learning instances.
2. While the results of the proposed method are highed than baselines, the successful rate is still not statisfactory, especially the result of `full-data' is still low. More discussion on the results and illustration of failure cases can be added.
3. The assumption of the paper that similar geometry indicate similar affordance for novel categories may be not solid.

**Questions:**

1. The `semantic' used in this paper is ambigious. You may add more explaination or use another term.
2. The similarity learning part needs more details. How do you use Eq(1) to learn the similairty?
3. How do you decide to stop the few-shot training?
4. What is sliding joint in L235?

**Limitations:**

The limiation of low success rate is not discurssed.

---

> ### Author Rebuttal · Authors · 2023-08-09
>
> We sincerely thank you for your efforts and your thoughtful feedback. Thank you for the valuable questions and suggestions!
> Here, we will address the questions and comments in the same order as they appear in the reviews. Further questions are welcomed!
>
>
> > Statistics, eg. std, are not provided in the experimental results.
>
> The detailed results (144 items) of different kinds of category combinations are shown in Table 1 in the supplementary material,
> different combination shows consistent results, which demonstrate the performance of our method is not affected by the choice of few-shot learning instrances.
>
> > While the results of the proposed method are higher than baselines, the successful rate is still not satisfactory.
>
> We follow Where2Act [1] and choose the sample successful rate to evaluate our prediction because it balances the precision and recall of successful predictions, which aims to evaluate coverage along with accuracy.
>
> > More discussion on failure cases and limitations.
>
> We add more failure cases analysis. For example, the similarity module may fail on categories that have significant shape variance (especially if the quality of the point cloud is low), which is shown in the PDF attached. When the size of uncertain geometry is very small, similarity might fail to capture the geometric information. **Please see the global reply for detailed discussion.**
>
> >  The assumption of the paper that similar geometry indicates similar affordance for novel categories may be not solid.
>
> Thanks for your thoughtful comment! Our similarity does not only depend on geometries but also on global features and action directions. As shown in the supplementary, the surface of a door is predicted to be similar to the front surface of a microwave but is different to the top surface of the microwave even if they are both flat surfaces. Moreover, our few-shot learning could further explore whether the geometry is similar through further interactions. We are happy to discuss more if you have more questions.
>
> > The 'semantic' used in this paper is ambiguous. You may add more explanation or use another term.
>
> 'Semantic' is used to clarify that the similarity here is not only based on geometries, but also based on the given action type (push or pull), which contains manipulation semantic information for different actions. For example, handles are significant in pulling whereas less important in pushing, and a horizontal handle could not be grasped by a gripper whose pose is also horizontal. We will add more explanations to make it more clear or change the term. Thanks for pointing it out!
>
> > The similarity learning part needs more details. How do you use Eq(1) to learn the similarity?
>
> The Eq(1) defines the supervision of the similarity module. Specifically, given a partial observation $O_i$, point to interact $p_i$ and the action direction $R_i$, the similarity is defined as an estimation of the accuracy of affordance prediction on this input $Aff(Oi,pi,Ri)$. The accuracy is computed by the correctness of affordance prediction during training $Accu(Aff(Oi,pi,Ri))$. The module is supervised using a L1 loss as described in Sec. 4.4. Any further questions are welcomed!
>
> > How do you decide to stop the few-shot training?
>
> As described in line 43 of the supplementary, the exploration will stop if the average score of similarity prediction exceeds a bar (0.9) or the interaction budget is reached.
>
> > What is sliding joint in L235?
>
> Sliding joint is the prismatic joint, thanks for pointing out this typo!
>
> [1] Kaichun Mo, Leonidas J Guibas, Mustafa Mukadam, Abhinav Gupta, and Shubham Tulsiani.
> Where2act: From pixels to actions for articulated 3d objects. ICCV 2021.

---

### Official Review · Reviewer_2ur3 · 2023-07-03

**Soundness:** 3 good
**Presentation:** 3 good
**Contribution:** 3 good
**Rating:** 6
**Confidence:** 4

**Summary:**

The paper focuses on the task of predicting affordances given a single-frame observation of an articulated object. The authors utilize few-shot learning to address the novel object adaption problem. However, previous approaches to few-shot learning require per-instance interaction during test time, which can be both costly and inefficient. To address this issue, this paper introduces the Where2Explore framework, which estimates geometric similarity across categories to offer more efficient interaction proposals. Additionally, it enables the transfer of learned affordance to novel cases. Experimental results demonstrate that this method surpasses previous approaches in various aspects.

**Strengths:**

* The motivation to explore more efficient few-shot learning is intriguing. The concept of leveraging affordance similarities across categories is intuitive and seems to be effective based on the experimental results.
* The training strategy to learn the cross-category geometry-aware affordance similarity is straightforward and effective based on the quantitative visualization.
* This paper conducts extensive experiments to compare with prior work and includes sensible ablation studies to demonstrate the effectiveness of the Where2Explore framework.

**Weaknesses:**

* There are not enough descriptions of the hard-coded intersection sets. Even if it’s proposed in the prior work, it’s better to include more details to be self-contained in this paper. This paper only chooses two actions, “push” and “pull” from the original 6 actions defined in the prior work. For the interaction definition, the action direction should be in the positive hemisphere of the surface normal. It’s not clear if it holds true for the real data. (Because the real data similarity visualization is not based on the surface normal)
* In the few-shot adaption stage, it’s unclear how many interactions for each point are sampled for the similarity. In addition, the authors mention the similarity is conditioned on the action. It’s unclear if the action is still conditioned in the object/camera/world coordinate or some local coordinate to the interaction points. If the action is still parameterized in the global coordinate, it’s hard to see if the model can really learn the similarity.
* For the few-shot learning, it’s hard to see the usefulness of the models in the supporting set. The current set of experiments cannot show the usefulness of the supporting set. Is this possible to learn from scratch with the few-shot learning strategy?
* For the affordance and similarity training, there seems no reason to train similarity at the start, because the affordance predictions are still quite bad. It’s also not clear if training affordance and similarity in two-stage can achieve better performance.

**Questions:**

* Why for the visualizations of the similarity prediction in figure 5, the action directions are different for synthetic data and the real data. Does the actual inference process have any difference for synthetic data and real data? For example, the sampled actions for few-shot learning are defined differently for the synthetic and the real data?
* For the interaction budget, it seems that 1, 2, 5 get better and better performance, is there some performance top bar, and what’s the number of the budgets?
* In the similarity visualization, why the dark blue is not considered as potential interactions (they should have the lowest similarity score?) Like the pot in figure 5 (the second row). It seems that the visualization mix the affordance and the similarity, which is a bit confusing.

**Limitations:**

The authors mention the limitation.

---

> ### Author Rebuttal · Authors · 2023-08-09
>
> We sincerely thank you for your efforts and your thoughtful feedback. Thank you for the valuable questions and suggestions!
> Here, we will address the questions and comments in the same order as they appear in the reviews.
>
> > There are not enough descriptions of the hard-coded intersection sets. This paper only chooses two actions, “push” and “pull” from the original 6 actions defined in the prior work. For the interaction definition, the action direction should be in the positive hemisphere of the surface normal. It’s not clear if it holds true for the real data.
>
> Thanks for your advice. For pushing, a closed gripper first touches the surface and then pushes 0.05 unit-length forward. For pulling, an open gripper approaches the surface by moving forward for 0.04 unit-length, performs grasping by closing the gripper, and pulls backward for 0.05 unit-length. We will add more detail about the action sets in the final paper.
>
> As for the other action primitives, we also provide experiments on two new action primitives: dragging (pulling object parts to the left) and scratching (pushing object parts to the left side using friction). The results show that our method still outperforms other method, indicating that our system has potential to be applied to more actions. **Please see the global reply for detailed results**
>
> For the action direction issue, the affordance of actions with directions in the negative hemispheres are supervised by zero during training. Therefore, in the real scenario, our model can distinguish the actions that are in the negative hemisphere and give negative affordance predictions.
>
> > It’s unclear how many interactions for each point are sampled for the similarity. It’s unclear if the action is still conditioned in the object/camera/world coordinate or some local coordinate to the interaction points.
>
> Thanks for pointing this out! We randomly sample ten directions for the whole objects and let the similarity module choose the point with lowest prediction. We will clarify it in the paper. As for the action coordinate issue, we only parameterize actions’ directions and gripper orientations. The starting point and ending point are hardcoded regarding the interaction point (e.g. pushing is defined as a closed gripper first touches the point and then pushes 0.05 unit-length forward). Therefore, our model could capture the local similarity conditioned on actions.
>
> > For the few-shot learning, it’s hard to see the usefulness of the models in the supporting set. Is this possible to learn from scratch with the few-shot learning strategy?
>
> Supporting set provides the affordance module with manipulation knowledge in one category, only with which the transfer of knowledge using the similarity is possible. Besides, the similarity is defined as similarity between geometries in the supporting set and those in novel categories. If no supporting set is provided, both the manipulation knowledge and the exploration strategy can not be learned.
>
> > For the affordance and similarity training, there seems no reason to train similarity at the start.
>
> Thanks for your valuable advice! We conduct more experiments on the two-stage training strategy. We observe no evident performance improvements (as shown below). Our observation is that affordance prediction will be stable and reasonable after about 10 epochs
> in the original training. We choose our current training strategy because it is easier to train and more efficient during few-shot learning, as the two modules share the same encoder which can be trained in two modules simultaneously.
>
> | F-score | pushing | pulling |
> |--|--|-|
> |Ours (One-stage) | 41.6%| 24.2% |
> |Two-stages| 42.4% | 22.3% |
>
> > Why for the visualizations of the similarity prediction in figure 5, the action directions are different for synthetic data and the real data. Does the actual inference process have any difference for synthetic data and real data?
>
> The inference process for real data and synthetic data are the same. We visualize differently simply because it is hard to obtain precise normal directions from the real data, so we choose to use "up" direction in the real-world visualization. However, note that our model is capable of predicting actions with any given directions.
>
> > For the interaction budget, it seems that 1, 2, 5 get better and better performance, is there some performance top bar, and what’s the number of the budgets?
>
> The number of the budgets means the number of interactions on each instance. As shown in the ablation study (line 302), "full-data" could be seen as the top bar of the performance since it is trained with abundant data (100 interactions on each shape). The results show using 5 as the interaction budget can achieve comparable performance of the top bar. We also tried interaction budget 10 and the performance did not show an increase (less than 3% improvement in both pushing and pulling). We will clarify it in the paper.
>
> > In the similarity visualization, why the dark blue is not considered as potential interactions?
>
> Following the setting of Where2Act, VAT-Mart, DualAfford [1, 2, 3], points that cannot result in the movement of an articulated part (i.e., the points that can only lead to the movement of the whole object) are segmented out, resulting in dark blue. Thanks for pointing this out! We will clarify it in the paper.
>
> [1] Kaichun Mo, Leonidas J Guibas, Mustafa Mukadam, Abhinav Gupta, and Shubham Tulsiani.
> Where2act: From pixels to actions for articulated 3d objects. ICCV 2021.
>
> [2] Ruihai Wu, Yan Zhao, Kaichun Mo, Zizheng Guo, Yian Wang, Tianhao Wu, Qingnan Fan, Xuelin Chen, Leonidas Guibas, and Hao Dong.
> Vat-mart: Learning visual action trajectory proposals for manipulating 3d articulated objects. ICLR 2022.
>
> [3] Yan Zhao, Ruihai Wu, Zhehuan Chen, Yourong Zhang, Qingnan Fan, Kaichun Mo, and Hao Dong. Dualafford: Learning collaborative
> visual affordance for dual-gripper object manipulation. ICLR 2023.

---

> > ### Comment · Reviewer_2ur3 · 2023-08-18
> >
> > Thanks for the responses from the authors. The responses have resolved all my questions.

---

### Official Review · Reviewer_WVQD · 2023-07-05

**Soundness:** 4 excellent
**Presentation:** 4 excellent
**Contribution:** 3 good
**Rating:** 6
**Confidence:** 4

**Summary:**

This paper studies the problem of interacting with novel articulated objects. More specifically, a few-shot learning based approach is proposed to adapt to unseen instances at test time. The methodology explicitly leverages similar local geometries, which may be shared across objects from distinct categories. Experimentation reveals superior performance relative to prior work.

**Strengths:**

- Well-written. The paper is well-written, and the figures aid in the understanding of the methodology.
- Ablations are insightful. The ablations adequately test the utility of the design choices of the proposed methodology.
- Strong results. The proposed approach outperforms baselines from prior work as well as the ablations. The fact that the proposed methodology achieves comparable performance to full-data while using only 0.3% of the data is impressive.

**Weaknesses:**

- Simplistic problem setting. The problem setting seems to be simplistic. For instance, the robot arm is abstracted away. This completely side-steps the problem of finding a motion plan which actually solves the task in a collision-free manner, limiting real world deployment. Furthermore, the input point cloud is assumed to be cleanly segmented out. It is unclear whether this is a realistic assumption, especially for when the models are deployed to the real world.
- Limited real world evaluation. It is not clear whether the model truly generalizes to the real world. Furthermore, the section on real world experimentation (both in the main paper and the supplementary) seems to be quite limited. For instance, how is the grasping (gripper open / close) done, as this doesn't seem to be an output of the system? Or, how is the motion plan obtained from the end-effector poses? More extensive (quantitative and qualitative) experimentation on real world data would be preferred to be convincing that the sim2real generalizes.

**Questions:**

- Are "action directions / gripper orientations" often the normals? Is it possible to constrain the action space to just be normals?
- Is a separate affordance model trained for each category, such that the similarity model can be trained on "held-out" categories?
- Other alternatives / potential ablations:
  - As an alternative to the L1 loss for training the similarity module, could a cross-entropy loss have been used for whether the current affordance model was correct on the unseen object? This would also provide a notion of uncertainty of the model (via confidence), which could be used to guide exploration.
  - What if you returned the output of the affordance model for points with the highest similarity, without any further exploration? Wouldn't this correspond to points for which the model is the most confident in being correct?
  - Could you train an affordance model which only operates on smaller point clouds (ie smaller field of view, e.g. a patch)? This may force the model to only look at the local geometries which should be shared across categories?
- "During few-shot learning, Where2Act will sample interaction on the points with an affordance score closest to 0.5." --> Why is this the case? Isn't 0 push and 1 pull? So shouldn't the highest confident action be sampled (ie farthest away from 0.5)?

**Limitations:**

There is a small section on limitations of the work, which could be expanded.

---

> ### Author Rebuttal · Authors · 2023-08-09
>
> We sincerely thank you for your efforts and your feedback. Thank you for the valuable questions!
> Here, we will address the questions and comments in the same order as they appear in the reviews.
>
> > The robot arm is abstracted away. The input point cloud is assumed to be cleanly segmented out.
>
> We follow the settings of a series of manipulation affordance works like Where2Act and VAT-Mart [1, 2] and focus on learning an object-centric visual prior that could guide the downstream tasks. Therefore we abstract away the robot arm to learn all possible interactions on each object. Current control systems (e.g. Inverse kinematics solver [3]) of robotic arms could achieve motion planning given the position of the end-effector, which facilitates our real-world application. It is also proved by our real-world experiments, where we use impedance control [4] to plan for the arm motion. As for the segmentation issue, since our model consumes partial point clouds following Where2Act and other works, segmentation could be easily performed by using the segmentation mask on rgb images and applying it to the depth maps. We can use SAM [5] to produce such segmentation masks in the real world.
>
> > Limited real world evaluation. How is the grasping (gripper open / close) done? How is the motion plan obtained from the end-effector poses?
>
> Thanks for your suggestions! We provide more real-world experiments in the global reply, which includes visualization of affordance and similarity prediction on novel categories (e.g. bucket, kettle and pot). The results demonstrate the generalization ability of our method in real scenarios. **Please see the global reply for detailed results.**
> The open and close of the gripper is controlled by hard-coded interaction sets defined in Where2Act. The degree of closing is controlled by the force sensor at the end-effector. When the pressure on the gripper is higher than a threshold, the gripper will stop closing. We use Impedance control system [4] for the real-world experiments. Given the poses of the end-effectors, the system could plan the motion trajectory. We will release the code and add more details in the final paper.
>
> > Are "action directions / gripper orientations" often the normals? Is it possible to constrain the action space to just be normals?
>
> Thanks for your nice question! The action directions refer to the motion directions and the gripper orientations are the poses of the gripper. We do not limit them to the normals since we want a dense prior that is conditioned on specific action parameters. This design enjoys many benefits. For example, a horizontal handle should be grasped by a gripper whose pose is vertical. When pushing a window that moves prismatically, the direction of action should also not be constrained to the normal to make it move.
>
> > Is a separate affordance model trained for each category, such that the similarity model can be trained on "held-out" categories?
>
> As we want one affordance model to learn all categories in the end (which is more practical in the real-world application), we perform few-shot learning on "held-out" categories by adapting the same model under the guidance of the similarity module.
>
> > As an alternative to the L1 loss for training the similarity module, could a cross-entropy loss have been used?
>
> As accuracy is a continuous number in [0, 1], we want to learn a continuous indicator of the uncertainty, thus precisely revealing the degree of uncertainty and finding the geometries with highest uncertainty to interact with. A cross-entropy loss might not guarantee this property. Previous work (Confidence-Aware Learning for Deep Neural Networks [6]) also proves that L1 is better in making the predicted numbers more smooth, meaningful and continuous than CE loss.
>
> > What if you returned the output of the affordance model for points with the highest similarity, without any further exploration?
>
> Following Where2Act, we aim to learn the dense pixel to actions mapping instead of just performing a task, which means we need to know which action could work as well as which couldn’t. Such visual priors will be helpful in diverse tasks. Besides, we want to learn affordance on novel geometries when faced with a novel category (e.g. handles of a mug), which requires exploration instead of just choosing the affordance with highest similarity.
>
> > Could you train an affordance model which only operates on smaller point clouds?
>
> Some kinematic information could only be discovered when seeing the global shapes (i.e. Whether the joint is at limit, whether the surface moves horizontally or vertically). Our network backbone (PointNet++) leverages a hierarchical structure to extract features at various scales, with each point having features containing both its local and global information, which benefits the prediction accuracy as well as the generalization ability.
>
> > Why does Where2Act sample interaction on the points with an affordance score closest to 0.5?
>
> We train separate models for different action primitives (e.g. pushing and pulling). In each primitive, the affordance
> prediction indicates whether one action will be successful (0 for fail and 1 for success). Seeing from a classification aspect, 0.5 indicates uncertainty.
>
> > Limitations could be expanded.
>
> Thanks for your valuable suggestion! We add more failure cases and limitation analysis. **Please see the global reply for more information.**
>
> [1] Where2act: From pixels to actions for articulated 3d objects. ICCV 2021
>
> [2] Vat-mart: Learning visual action trajectory proposals for manipulating 3d articulated objects. ICLR 2022
>
> [3] Wikipedia contributors. "Inverse kinematics." Wikipedia, The Free Encyclopedia. Wikipedia, The Free Encyclopedia
>
> [4] Wikipedia contributors. "Impedance control." Wikipedia, The Free Encyclopedia. Wikipedia, The Free Encyclopedia
>
> [5] Segment anything. arXiv preprint arXiv:2304.02643. 2023
>
> [6] Confidence-aware learning for deep neural networks. ICML 2020

---

> > ### Comment · Reviewer_WVQD · 2023-08-21
> >
> > Thanks for the response. The authors' clarifications have addressed my questions.

---

### Official Review · Reviewer_8WHp · 2023-07-07

**Soundness:** 3 good
**Presentation:** 3 good
**Contribution:** 3 good
**Rating:** 5
**Confidence:** 5

**Summary:**

The paper focuses on few-shot affordance learning and articulated object manipulation. It introduces a framework named Where2Explore, which consists of the affordance module and the similarity module. Both modules share the same architecture, a Where2Act [24]-like network based on PointNet++ [28]. While the affordance module is trained on a single category, the similarity model is trained across categories to understand similarity in local geometry and manipulation semantics. The model is benchmarked on 942 instances from 14 (3 for train, 11 for test) object categories in PartNet-Mobility [37]. It outperforms previous works [24, 34] in simulation.

**Strengths:**

1. The methodology is well motivated: Learning similar local geometry across object categories could benefit few-shot generalization in manipulation.
2. It is impressive to observe the stark differences between the seen categories and the unseen categories.
3. Under the same settings, Where2Explore exhibits a clear edge over previous works [24, 34] quantitatively.
4. The demo video is well-prepared.

**Weaknesses:**

1. The method is limited to pulling and pushing actions, and the action trajectories are short-term and hard-coded. These limitations inherited from Where2Act could result in a significant simulation-to-real gap.
2. There is only a single example of real-world manipulation, which is insufficient to demonstrate the method’s capabilities. Besides, according to the video, it seems that the gripper may not always hold mugs firmly enough. More real-world experiments should be conducted, and their success rates should be reported.
3. Few improvements on the architecture. The current module largely follows the design of [24, 34]. While it is commendable to demonstrate new insights using a minimal network design, the current task may benefit from more fine-grained cross-category transfer learning. There is room for improvement in the methods.
4. Neural Descriptor Fields (Simeonov *et al*.) is also capable of capturing local geometry information using few shots. It would be valuable to discuss and even adapt them for comparison.

**Questions:**

1. [38, 8] share a similar spirit and motivation in learning local geometry across categories and generalizing to the novel, which should be discussed in the related works.
2. Table 3: The successful rate of pulling drops for the full-data compared to ours. Any explanation for this observation?
3. Is the interaction budget (1, 2, 5) per instance or per category (10 instances)? This term seems not well defined.
4. L233: Which category is the affordance category ($C_{aff}$) in each combination?

**Limitations:**

The paper includes a section dedicated to limitations; however, it provides minimal discussion on the specific limitations. Please incorporate a comprehensive discussion of failure cases and limitations, including those in the Weaknesses section. To enhance the credibility of this work, it would be beneficial to incorporate quantitative results from real-world experiments and quantitative comparison to other works that follow a different paradigm, such as Neural Descriptor Fields and GAPartNet [8]. This would provide a more convincing evaluation of the proposed methodology.

---

> ### Author Rebuttal · Authors · 2023-08-09
>
> We sincerely thank you for your efforts and your thoughtful feedback. Thank you for the valuable questions and suggestions!
> Here, we will address the questions and comments in the same order as they appear in the reviews. Please feel free to follow up if you have more questions.
>
>
> > The method is limited to pulling and pushing actions, and the action trajectories are short-term and hard-coded. These limitations inherited from Where2Act could result in a significant simulation-to-real gap.
>
> Thanks for your nice comment! We use push and pull as they are the most commonly used action primitives in Where2Act and follow-up papers. Moreover, we have conducted two more experiments on two new action primitives following Where2Act: dragging (pulling object parts to the left) and scratching (pushing object parts to the left using friction). These action primitives are commonly used in the real world (e.g. scratch sliding windows to open them, dragging a laptop to close it). **Please refer to the global reply for detailed results.**
> As for the short-term issue, we follow the setting of Where2Act that aims to learn an object-centric dense visual prior. Although they are learned based on short-term interactions, such dense prior could guide the first step of long-term tasks. Current works like VAT-Mart[1] and UMPNet[2] have leveraged this kind of prior to perform long-term manipulations, which proves that the learned knowledge could help the downstream tasks.
>
>
> > There is only a single example of real-world manipulation.
> > Besides, according to the video, it seems that the gripper may not always hold mugs firmly enough. More real-world experiments should be conducted, and their success rates should be reported.
>
> We conduct more real-world experiments, which include affordance and similarity prediction on more novel categories (e.g. bucket, kettle and pot). **Please see the global reply for detailed results.**
> The problem that the gripper may not always hold mugs firmly enough comes from the small surface-friction between the gripper and the mug, and changing the gripper type could solve this problem.  Thanks for bringing up this point and we will make this clear in the final paper.
>
> > Few improvements on the architecture.
>
> Thanks for this comment! As you mentioned, we largely follow the network designs of previous works to demonstrate the new insights with a new learning system. We believe adding more designs to the network architectures is promising for future works to explore. We could let the model explicitly group the points into parts and reason the similarity among these parts across categories (e.g. the body part of a cup and a bottle is similar whereas the handle on the cup is unfamiliar). Such modification to the network might help the reasoning of similarity in part level.
>
> > Neural Descriptor Fields (Simeonov et al.) is also capable of capturing local geometry information using few shots.
> > It would be valuable to discuss and even adapt them for comparison.
>
> Thanks for bringing up this related work to us! We will certainly add discussion to this paper in the final revision. Neural Descriptor Fields leverage correlation among different instances in the same category to perform category-level few-shot imitation learning whereas our method focuses on cross-category few-shot learning. Specifically, NDF learns from the demonstration on a mug and transfers the knowledge to unseen mugs. We learn affordance on cabinets and windows and transfer to novel categories like tables and mugs with 5 interactions.
>
> > [38, 8] share a similar spirit and motivation in learning local geometry across categories and generalizing to the novel, which should be discussed in the related works.
>
> Thanks for your suggestion! Gapartnet and PartAfford both propose novel supervisions (part-level annotations or sparse labels) on parts to learn the affordance that can generalize to unseen categories, which are very interesting and useful. Different from these works, we do not receive explicit part-level supervisions, instead, our model learns similarity among local geometries through interaction results (i.e. parts are learned by the network itself). We will discuss more about these works in our related works.
>
> > The successful rate of pulling drops for the full-data compared to ours. Any explanation for this observation?
>
> We observe that models trained with abundant data could learn some corner cases (e.g., pulling the drawer inner surface by friction to make it open), which are not always successful, but included in the dataset as long as the number of random interactions is large enough. In contrast, our method with self-proposed interactions tends to explore geometries that are more meaningful (e.g. the handle of the drawer), thus achieving even higher performance in the successful rate.
>
> > Is the interaction budget (1, 2, 5) per instance or per category (10 instances)? This term seems not well defined.
>
> The interaction budget refers to the per-instance budget here. We will make it clearer in the paper. Thanks for pointing this out!
>
> > Which category is the affordance category in each combination?
>
> The affordance category is the first one of each category. We will clarify it in the paper.
>
> > More specific failure cases and limitations
>
> Thanks for your valuable suggestion. We will add more failure cases analysis. For example, the similarity module may fail on categories that have significant shape variance, which is shown in the PDF attached. As for the limitation, while our method has dramatically improved the data efficiency, we still need abundant data in the supporting set to form a fine-grained manipulation prior before we can perform few-shot learning. **Please see the global reply for detailed descriptions.**
>
> [1] Vat-mart: Learning visual action trajectory proposals for manipulating 3d articulated objects. ICLR 2022.
>
> [2] Universal manipulation policy network for articulated objects. RA-L, 2022.

---

### Official Review · Reviewer_ztRo · 2023-07-07

**Soundness:** 3 good
**Presentation:** 3 good
**Contribution:** 3 good
**Rating:** 6
**Confidence:** 2

**Summary:**

The paper explores a few-shot learning strategy to predict the affordance of objects in novel categories. The key idea of the paper is to use a similarity module to transfer the knowledge from known categories to novel categories. Then the affordance prediction on the novel object can be learned with just a few exploration actions. The paper is compared with several existing methods on standard benchmarks.

**Strengths:**

The paper addresses an important and interesting problem of transferring affordance knowledge from known categories to novel categories.

The paper proposes a similarity module for effective affordance knowledge transfer and achieves quantitativel  better results than existing methods.

The paper is overall easy to follow.


**Weaknesses:**

Method:

I have a couple of questions regarding the learning procedure and intuition.

1. In Figure 2, to train the similarity module, the affordance module predicts the affordance map on the novel object and the similarity module predicts the per-point similarity. In my understanding, is the similarity module exactly the same with the affordance module but just learned with different data and learning objectives? If so, in eq (1), is the Aff the affordance module or similarity module in Figure 2? I am not sure if eq (1) and Figure 2 are aligned, maybe explaining in more detail could be helpful.

2. It is not clear to me how the affordance prediction is updated during exploration/inference. Is it more of a test-time-adaptation? What module parameters are updated?

3. I am not sure if the intuition of generalization using local geometries is well supported by the design. Is the similarity module responsible for this? If so, how do local geometries play a part? Isn’t it still predicted for all N observed points with global information? Since this builds the key idea of cross-category generalization, I think further clarification is necessary.

4. In the paper, 10 instances from each novel category are used for few-shot learning. I wonder, for example, for Where2Act baseline, is it also trained with only 10 instances? There is also no ablation on the number of instances required, could it be fewer?

5. The paper only explores push and pull. Maybe more actions can be studied to validate the method.


**Questions:**

I think overall the paper addressed an interesting problem and proposed an effective solution, but I think further clarification on the method is necessary. Please see my comments above.

**Limitations:**

Yes.

---

> ### Author Rebuttal · Authors · 2023-08-09
>
> We sincerely thank you for your time and your positive feedback. Thank you for the valuable questions! Here, we will address the raised questions and comments in the same order as they appear in the reviews.  Please Feel free to follow up if you have more questions.
>
> > is the similarity module exactly the same with the affordance module but just learned with different data and learning objectives? If so, in eq (1), is the Aff the affordance module or similarity module in Figure 2?
>
>   Sorry for not being clear. We will make this clearer in the final paper.
> The two modules share the same encoder but have separate decoders. Their outputs are also defined differently. The Eq(1) defines the supervision of the similarity module, which is the accuracy of the prediction from the affordance module. The "Aff" in Eq(1) refers to the affordance module in Figure 3 (top). As shown in figure 3, the similarity module learns from
> the comparison of affordance prediction and ground truth (bottom right), which correspond to $Accu(Aff(O_i,p_i,R_i),m_i)$ in Eq(1). As for the Figure 2, it describes the general pipeline of our system, which is training affordance module on supporting set (left) and learning similarity through cross-category comparison (middle) and then perform few-shot learning under the guidance of the similarity module (right). Any further questions are welcomed!
>
> > how the affordance prediction is updated during exploration/inference?
>
> As written in line 187, both the affordance module and the similarity module are adapted using the proposed interactions and their corresponding results. The parameters of the affordance module are updated by the interaction results (successful or not) same as the training process. After the affordance module is updated, the similarity module will be supervised by the accuracy of affordance prediction on these interactions.
>
> > I am not sure if the intuition of generalization using local geometries is well supported by the design.
>
> Thanks for your insightful comment! While we don’t explicitly measure the similarity of local geometries, our architecture and the training method exploits this intuition in an implicit yet effective way. Firstly, the backbone (PointNet++) of the similarity module does not only depend on global features. PointNet++ leverages a hierarchical structure to extract features at various scales, with each point having features containing both its local and global information, and thus is able to capture local geometric information. Such ability is demonstrated in other affordance learning works that require learning local geometric information (Where2Act, DualAfford [1, 2]). Second, we train the similarity module in a cross-category manner, and the optimization target implicitly trains the network to focus on local geometries (i.e., recognizing known and unfamiliar local geometries) in order to achieve better generalization towards unseen categories. Finally, qualitative results empirically show that the learned similarity has a clear focus on local geometries. For example, in figure 5, the pot lid (middle) and the top surface of drawers (top) are predicted to be unfamiliar, which shows a clear geometric boundary with parts that the model is familiar with.
>
> > Is the Where2Act baseline also trained with only 10 instances?
>
> Yes. Data setting is the same for all baseline comparisons for a fair evaluation. Additionally, we compare our method with the original Where2Act that is trained on abundant data ('full-data' in Table 3). our method achieves comparable performance with only 0.3% data of the "full-data" model.
>
> > There is also no ablation on the number of instances required, could it be fewer?
>
> Thanks for your advice! We have conducted two more experiments respectively with 2 instance and 5 instances, the results are shown here. We could see the performance slightly drop for pushing and drop dramatically for pulling due to fewer positive supervision and instance-level variance.
>
> |   F-score   | pushing | pulling |
> |-------------|---------|---------|
> | Ours (10 instances)| 41.6%   | 24.2%   |
> | 5 instances | 40.3%   | 14.5%   |
> | 2 instances | 33.9%   | 12.8%   |
>
> > The paper only explores push and pull. Maybe more actions can be studied to validate the method.
>
> Thanks for your advice! We use push and pull as they are the most commonly used action primitives in Where2Act and follow-up papers [1, 2, 3]. Following Where2Act, we focus on learning dense visual prior for short-term manipulations and most of the short-term actions start with push or pull.
>
> Moreover, we have conducted two more experiments on two new action primitives following Where2Act: dragging (pulling object parts to the left) and scratch (pushing object parts to the left side using friction). These action primitives are commonly used in the real world (e.g. scratch sliding windows to open them, dragging a laptop to close it). **Please see the global reply for detailed results**. We could see our method still outperforms the baselines. Any further suggestions are welcomed!
>
> [1] Kaichun Mo, Leonidas J Guibas, Mustafa Mukadam, Abhinav Gupta, and Shubham Tulsiani.
> Where2act: From pixels to actions for articulated 3d objects. ICCV 2021.
>
> [2] Yan Zhao, Ruihai Wu, Zhehuan Chen, Yourong Zhang, Qingnan Fan, Kaichun Mo, and Hao Dong. Dualafford: Learning collaborative
> visual affordance for dual-gripper object manipulation. ICLR 2023.
>
> [3] Wang, Y., Wu, R., Mo, K., Ke, J., Fan, Q., Guibas, L. J., & Dong, H. Adaafford: Learning to adapt manipulation affordance for 3d articulated objects via few-shot interactions. ECCV 2022.

---

> > ### Comment · Reviewer_ztRo · 2023-08-20
> >
> > Thanks for the feedback. The authors' clarifications addressed my questions.

---

### Author Rebuttal · Authors · 2023-08-09

We thank you all for your efforts and insightful comments! We thank reviewers for appreciating our motivation: "The methodology is well motivated", "The motivation to explore more efficient few-shot learning is intriguing", and design: "The concept of leveraging affordance similarities across categories is intuitive", "The paper proposes a similarity module for effective affordance knowledge transfer".
Below we clarify some common concerns. Thanks again for valuable comments! Any further questions are welcomed!

> More real-world evaluations

We conduct more real-world experiments on diverse objects, which includes affordance and similarity prediction on more novel categories (10 instances including buckets, pots and kettles). We choose the vertically pulling up as our action for the real-world experiments and give 5 interaction budget for each instance. While our model is only trained on cabinet, window and faucet and the affordance module fails to give reasonable predictions, the result indicates that the similarity module could still provide reasonable predictions on novel objects in the real world, guiding the exploration of uncertain yet important geometries. After the exploration, the adapted affordance could reveal the dense kinematic information on the novel objects. **Please see the PDF attached for detailed visualization.**

> More action primitives

We use push and pull as they are the most commonly used action primitives in Where2Act [1] and follow-up papers. Moreover, we have conducted two more experiments on two new action primitives following Where2Act: dragging (pulling object parts to the left) and scratching (pushing object parts to the left side using friction). These action primitives are commonly used in the real world (e.g. scratch sliding windows to open them, dragging a laptop to close it). We could see our method still outperforms previous baselines (especially in dragging, which is more demanding). **Please see the PDF attached for detailed results.**

> More discussion on failure cases and limitations

We add more failure cases analysis. For example, the similarity module may fail on categories that have significant shape variance (especially if the quality of the point cloud is low), which is shown in the PDF attached. Besides, when the size of uncertain geometry is very small, the similarity module might fail to capture the geometric information. **Please see the PDF attached for detailed visualization.**
As for the limitation, while our method has dramatically improved the data efficiency, we still need abundant data in the supporting set to form a fine-grained manipulation prior before we can perform few-shot learning. Besides, in order to extend our method to long-term manipulations, we might need to rely on some recent works such as VAT-Mart [2] and UMP-Net [3].

[1] Kaichun Mo, Leonidas J Guibas, Mustafa Mukadam, Abhinav Gupta, and Shubham Tulsiani.
Where2act: From pixels to actions for articulated 3d objects. ICCV 2021.

[2] Ruihai Wu, Yan Zhao, Kaichun Mo, Zizheng Guo, Yian Wang, Tianhao Wu, Qingnan Fan, Xuelin Chen, Leonidas Guibas, and Hao Dong.
Vat-mart: Learning visual action trajectory proposals for manipulating 3d articulated objects. ICLR 2022.

[3] Xu Z, He Z, Song S. Universal manipulation policy network for articulated objects[J]. IEEE Robotics and Automation Letters, 2022.

---

### Decision · Program_Chairs · 2023-09-21

**Decision:**

Accept (poster)

**Comment:**

This paper studies a novel approach on affordance with articulated objects. The goal is aiming at generalization to novel categories. All the reviewers agree to accept the paper after the rebuttal and the AC agrees.